# Magnesium Isoglycyrrhizinate Reduces the Target-Binding Amount of Cisplatin to Mitochondrial DNA and Renal Injury through SIRT3

**DOI:** 10.3390/ijms232113093

**Published:** 2022-10-28

**Authors:** Xinyu Wang, Hutailong Zhu, Jiayin Hu, Haobin Li, Suhan Guo, Bin Chen, Changxiao Liu, Guangji Wang, Fang Zhou

**Affiliations:** 1Key Laboratory of Drug Metabolism and Pharmacokinetics, Haihe Laboratory of Cell Ecosystem, China Pharmaceutical University, Nanjing 210009, China; 2Institute of Clinical Pharmacology & Department of Pharmacy, General Hospital of Ningxia Medical University, Yinchuan 750004, China; 3State Key Laboratory of Drug Delivery Technology and Pharmacokinetics, Tianjin Institute of Pharmaceutical Research, Tianjin 300193, China

**Keywords:** cisplatin, magnesium isoglycyrrhizinate, renal injury, SIRT3, mitochondria, mtDNA

## Abstract

Nephrotoxicity is the dose-limiting factor of cisplatin treatment. Magnesium isoglycyrrhizinate (MgIG) has been reported to ameliorate renal ischemia–reperfusion injury. This study aimed to investigate the protective effect and possible mechanisms of MgIG against cisplatin-induced nephrotoxicity from the perspective of cellular pharmacokinetics. We found that cisplatin predominantly accumulated in mitochondria of renal tubular epithelial cells, and the amount of binding with mitochondrial DNA (mtDNA) was more than twice that with nuclear DNA (nDNA). MgIG significantly lowered the accumulation of cisplatin in mitochondria and, in particular, the degree of target-binding to mtDNA. MgIG notably ameliorated cisplatin-induced changes in mitochondrial membrane potential, morphology, function, and cell viability, while the magnesium donor drugs failed to work. In a mouse model, MgIG significantly alleviated cisplatin-caused renal dysfunction, pathological changes of renal tubules, mitochondrial ultrastructure variations, and disturbed energy metabolism. Both in vitro and in vivo data showed that MgIG recovered the reduction of NAD^+^-related substances and NAD^+^-dependent deacetylase sirtuin-3 (SIRT3) level caused by cisplatin. Furthermore, SIRT3 knockdown weakened the protective effect of MgIG on mitochondria, while SIRT3 agonist protected HK-2 cells from cisplatin and specifically reduced platinum-binding activity with mtDNA. In conclusion, MgIG reduces the target-binding amount of platinum to mtDNA and exerts a protective effect on cisplatin-induced renal injury through SIRT3, which may provide a new strategy for the treatment of cisplatin-induced nephrotoxicity.

## 1. Introduction

As cisplatin is an effective antitumor agent, cisplatin-based combination chemotherapy is widely used as a first-line option for solid tumors [1]. In clinical practice, patients primarily receive multiple doses administered weekly or monthly, and the efficacy is dose-dependent, which is often accompanied by cumulative and dose-dependent nephrotoxicity [2]. Despite hydration, cisplatin is responsible for 20% of all acute renal injury (AKI) cases in hospitalized cancer patients [2]. AKI can increase the risk of death by 10- to 15-fold and result in a mortality rate of 50%. Severe and recurrent AKI is also an independent risk factor for chronic kidney disease (CKD) [3]. Several clinical studies [4,5,6] have shown that the use of cisplatin is associated with the incidence and progression of CKD in cancer survivors. Moreover, growing at a rate of 1% per year, CKD has become a global health burden [7]. Thus, there is an urgent need to develop new therapeutic approaches to prevent and treat platinum-induced nephrotoxicity.

The key molecular mechanisms involved in cisplatin-induced nephrotoxic adverse effects include cellular uptake and accumulation, inflammation, oxidative stress, vascular injury, endoplasmic reticulum stress, and necrosis and apoptosis [8]. Renal proximal tubules are the main targets for the damaging effects of cisplatin nephrotoxicity [9]. Cisplatin accumulates in the renal tubular epithelial cells, leading to mitochondrial structural damage and increasing steady-state levels of reactive oxygen species (ROS) [10]. There is increasing evidence [11,12,13] suggesting that mitochondrial dysfunction plays a crucial role in several kidney diseases. Generally, it is considered that cisplatin predominantly binds to nuclear DNA (nDNA). However, it was reported that in rats exposed to a single dose of cisplatin during gestation, cisplatin-DNA binding levels in liver and brain mitochondrial DNA (mtDNA) are higher than those observed in genomic DNA [14]. Whether cisplatin could directly bind to the mtDNA of renal tubular epithelial cells leading to renal mitochondrial dysfunction remains unknown. Here, we used inductively coupled plasma mass spectrometry (ICP-MS) to quantify the DNA-binding levels of platinum metal in subcellular organelles. 

Magnesium isoglycyrrhizinate (MgIG), a magnesium salt of 18α-glycyrrhizic acid stereoisomer, is widely used to treat inflammatory liver diseases because of its potent anti-inflammatory and hepatoprotective effects [15,16]. MgIG also improved mitochondrial microstructure and energy metabolism in the liver of mice [15]. Glycyrrhetinic acid is a metabolite of glycyrrhizic acid, which has been developed into mitochondrial-targeted drug delivery systems in some studies [17,18,19]. Therefore, it is speculated that MgIG may target mitochondria. Previous studies showed that MgIG improved renal injury caused by ischemia–reperfusion [20] or arsenic trioxide aminoglycosides [21]. However, it remains unknown whether MgIG has a protective effect on renal injury caused by cisplatin. Accordingly, we hypothesize that MgIG has a protective effect on cisplatin-induced renal injury by targeting mitochondria. To better mimic clinical situations, a low-dose repeated administration of cisplatin-induced kidney injury mouse model was employed in this study [22,23,24]. The present study aims to evaluate the protective effect of MgIG on cisplatin-induced nephrotoxicity and to explore the underlying mechanisms.

## 2. Results

### 2.1. MgIG Reduces the Concentration of Cisplatin in Mitochondria and Target-Binding Amount to mtDNA

First, we examined the subcellular distribution of platinum and MgIG in human renal tubular epithelial cells HK-2. The amount of platinum was higher in mitochondria than that in cytoplasm and nuclei (Figure 1A). Similarly, the amount of MgIG in mitochondria was approximately 4.7 times higher than that in nuclei (Figure 1B) but not significantly different from that in the cytoplasm. The concentration–time curves of platinum in mitochondria and nuclei showed a relatively rapid rise of platinum concentration in mitochondria and nuclei during the first 12 h and little change during the last 12 h (Figure 1C). From start to 4 h, no difference was observed between mitochondria and nuclei; from 6 h onward, the platinum concentration in mitochondria increased to 1.6–2.1 times the level in nuclei, suggesting that more platinum accumulated in mitochondria with time. After co-administration with MgIG, platinum was significantly reduced in mitochondria from 8 h to 24 h, and was also remarkably decreased in nuclei at 12 h and 24 h. At 24 h, MgIG caused a 47.3% reduction in platinum concentration in mitochondria, and a 37.0% reduction in nuclei (Figure 1D). Considering that the high affinity of platinum to DNA and the extent of platinum–DNA adduct formation are closely related to its toxicity, the effect of MgIG on the binding amount of platinum to mtDNA and nDNA was investigated. MgIG pointedly reduced the binding amount of platinum to mtDNA, while it failed to affect the level in platinum–nDNA adducts (Figure 1E).

Next, we established a cisplatin-induced renal injury model which mimicked the clinical chemotherapy cyclic administration by using a multi-cycle, repeated daily low-dose regimen (Figure 1F, 3.5 mg/kg daily, four consecutive days of intraperitoneal injection followed by 10 days of rest for three cycles). Combined administration of MgIG reduced the platinum concentration by 30.3% in mouse kidneys (Figure 1G). Consistent with the in vitro results, MgIG decreased the concentration of cisplatin in mitochondria (Figure 1H) and its binding amount to mtDNA (Figure 1I), but had no obvious effect on the binding amount of platinum to nDNA in mouse kidneys.

To explore the reasons for these findings, the mRNA or protein levels of related transporters governing the disposition of cisplatin in kidney was examined (Appendix A). The platinum uptake transporters-organic cation transporter 2 (OCT2) was not expressed in HK-2 cells, while the expression of copper transporter 1 (CTR1) did not differ between the cisplatin group and MgIG + cisplatin group. Cytochrome C oxidase copper chaperone (COX17), responsible for transporting cisplatin from the cytoplasm to mitochondria, also showed no significant changes. In the kidney tissue of cisplatin-induced renal injury mouse model, administration with or without MgIG did not affect the mRNA levels of OCT2, multidrug resistance protein 1 (MDR1) and multidrug and toxin extrusion 1 (MATE1) compared to the cisplatin group.

### 2.2. MgIG Significantly Ameliorates Cisplatin-Induced Renal Injury

Following this, the renal function of mice was investigated. It is shown that serum creatinine (SCr) and urea nitrogen (BUN) levels in the cisplatin group increased continuously throughout the administration period. At the end of cycle 3, SCr and BUN were 2.1-fold and 1.8-fold higher than those in the control group, respectively, indicating successful modeling. Compared with the cisplatin group, SCr levels in the MgIG + cisplatin group were 0.6-fold, 0.7-fold, and 0.6-fold (Figure 2A,B) lower in cycles 1, 2, 3, respectively. In addition, BUN exhibited a similar trend. After three cycles, mRNA levels of KIM-1 and NGAL in the kidneys, markers of renal tubular injury, decreased 0.52 times and 0.56 times after the combination of MgIG, respectively (Figure 2C,D).

Next, pathological changes in the kidneys were analyzed (Figure 2E), and the results of hematoxylin and eosin (H&E) staining revealed that cisplatin administration caused heavy inflammatory infiltration in the kidneys compared with the control group; however, this effect was mitigated by the combination with MgIG. Furthermore, we observed the ultrastructure of mitochondria in mouse renal tubules by projection electron microscopy. Mitochondrial cristae appeared to be fractured and translucent, and the stromal region became abnormally translucent in the cisplatin group, while the morphology of cristae normalized after MgIG combination.

### 2.3. MgIG Notably Alleviates Cisplatin-Induced Mitochondrial Dysfunction

Subsequently, we investigated whether MgIG alleviated cisplatin-induced mitochondrial dysfunction in HK-2 cells. Compared with control, the percentage of JC-1 monomer–positive cells increased from 3.6% ± 0.3% to 21.3% ± 0.8% after 20 μM cisplatin treatment of HK-2 cells for 24 h, and decreased to 8.7% ± 1.0% (cisplatin versus MgIG + cisplatin) after co-administration with MgIG. These data suggest that MgIG notably restores the cisplatin-induced depolarization of mitochondrial membrane potential in HK-2 cells (Figure 3A,B).

As mitochondria are the major source of intracellular ROS, the level of mitochondrial ROS was also examined. MitoSOX immunofluorescence results (Figure 3C,D) showed that the level of mitochondrial superoxide of HK-2 cells rose 1.5-fold (*p* < 0.001) relative to that of the control after cisplatin treatment, and only 1.2-fold relative to that of the control after the combination with MgIG, which was significantly different from that of the cisplatin group (*p* < 0.01), indicating that MgIG was clearly able to diminish the cisplatin-induced mitochondrial ROS production in HK-2 cells.

Mitochondrial morphology plays an essential role in maintaining cellular physiological functions. Thus, we used the MitoTracker^®^ Deep Red probe to display the morphology of mitochondria. The control group showed elongated tubular or continuous reticular structures, while cisplatin altered the mitochondrial morphology in HK-2 cells to a discontinuous short rod- or dot-like shape (Figure 3E). The average length was reduced by 0.6-fold relative to that of the control group; when co-treated with MgIG, the average length of mitochondria recovered to the normal level. Interestingly, MgIG alone increased the length of mitochondria 1.2-fold relative to that of the control. Hence, the combination with MgIG improved the negative effect of cisplatin on mitochondrial morphology and maintained mitochondrial integrity (Figure 3F).

The primary function of mitochondria is to complete electron transfer through the respiratory chain, followed by oxidative phosphorylation to generate ATP. Here, we measured mitochondrial oxygen consumption using the Seahorse Bioenergetics Assay. In comparison with the control, basal oxygen consumption was markedly lower; after administration of the uncoupling agent FCCP, the increase in mitochondrial oxygen consumption rate (OCR) was notably lower than that of the control, indicating a diminished maximal respiratory capacity. In contrast, the combination with MgIG reversed the cisplatin-induced changes in the OCR of HK-2 cells (Figure 3G). The oxygen consumption of basal respiration increased from 54.2 ± 10.6 pmol/min to 77.8 ± 6.3 pmol/min, and the oxygen consumption of maximal respiratory capacity increased from 110.5 ± 20.6 pmol/min to 145.4 ± 20.0 pmol/min (Figure 3H). Our results also showed that cisplatin decreased the concentration of intracellular ATP, and MgIG reversed this alteration (Figure 3I). In summary, MgIG ameliorates the mitochondrial dysfunction caused by cisplatin.

MgIG is a magnesium salt of 18α-glycyrrhetinic acid. It has been reported [25,26] that magnesium salts have a protective effect on renal tubular injury induced by cisplatin. To identify the key components that exert pharmacological effects, we also examined the impact of two magnesium salts (MgSO_4_ and MgCl_2_) with the same concentration as MgIG and found that they had no effect on the inhibition of HK-2 cell viability caused by cisplatin (Figure 3J).

### 2.4. MgIG Ameliorates Cisplatin-Induced Energy Metabolism Disorder and Recovers NAD^+^ Biosynthesis

To investigate the potential mechanism of MgIG protection, the metabolic patterns of endogenous small molecules in the kidneys were compared. The principal component analysis clearly distinguished between the groups (Figure 4A). The renal metabolic profile of the cisplatin group was conspicuously altered in comparison with that of the control group. In contrast, the renal metabolic profile of the MgIG + cisplatin group was close to the control group, suggesting that MgIG had a modifying influence on the abnormal renal metabolic profile caused by cisplatin.

Additionally, 52 compounds were screened using *p* < 0.05 as the criterion. MgIG exhibited marked modulation of reduced NAD^+^, adenosine, nicotinamide adenine dinucleotide phosphate (NADPH), and N-acetyl ornithine, and reduction of rising inosine diphosphate (IDP), allantoin, glutathione, and glucosamine (Figure 4B). KEGG pathway enrichment analysis (Figure 4C) indicated that MgIG regulated arginine biosynthesis, pentose phosphate pathway, nicotinic acid, and nicotinamide metabolism, as well as alanine, aspartate, and glutamate metabolism. Arginine biosynthesis, niacin and nicotinamide metabolism, and amino acid metabolism are closely related to mitochondrial function. In the kidneys, both niacin and nicotinamide are readily metabolized to NAD^+^, and a decrease in niacin and nicotinamide points to a disturbance in NAD^+^ metabolism. As previous studies have reported that NAD^+^ exhaustion is a defining event in the pathogenesis of AKI [27,28,29], NAD^+^ may become a hub connecting AKI to CKD [30,31] and plays an essential role in the development of CKD [32]. Therefore, we next focused on NAD^+^ metabolism and examined the levels of related substances.

The main biosynthesis pathways of NAD^+^ are shown in Figure 4D. Most NAD^+^ is salvaged from nicotinamide (NAM) or various forms of niacin taken up in the diet including NAM, NA, nicotinamide riboside (NR), and nicotinamide mononucleotide (NMN) [33]. In the kidneys of mice, NAD^+^, NAM, NMN, NR, and nicotinamide adenine dinucleotide phosphate (NADPH), but not nicotinamide adenine dinucleotide (NADH), were significantly downregulated by cisplatin treatment (Figure 4E). After the combined treatment with MgIG, we showed that NAD^+^, NMN, and NR were all back-regulated to different degrees. Likewise, a similar phenomenon was observed in HK-2 cells (Figure 4F). In summary, MgIG ameliorates cisplatin-induced energy metabolism disorder and recovers NAD^+^ biosynthesis.

### 2.5. MgIG Can Bind to SIRT3 and Prevent Cisplatin-Induced Decrease of SIRT3

Strategies for NAD^+^ supplementation have demonstrated outstanding potential, and one of the possible mechanisms is the activation of deacetylases (sirtuins) [34]. Mitochondria contain the largest intracellular pool of NAD^+^, and sirtuins in mitochondria have been shown to play an important role in renal protection. For these reasons, we examined the protein expression levels of sirtuins (SIRT3, SIRT4, SIRT5) located in mitochondria. When incubated with cisplatin, the expression levels of all three sirtuins were downregulated to different degrees in the kidneys, among which the level of SIRT3 decreased the most, measuring approximately 0.40 times the level of the control (Figure 5A–D). After the combination with MgIG, SIRT3 showed a back-regulation, rising 1.38 times higher than that of the cisplatin group. A similar phenomenon was observed in HK-2 cells (Figure 5E–H).

Subsequently, we subjected MgIG and SIRT3 to molecular interaction pattern analysis (Figure 5I,J). The results showed that MgIG could form hydrogen bonds with ALA at 146, ARG at 158, THR at 320, SER at 321, GLU at 323, and VAL at 324 of SIRT3, and π–π interaction with ARG at 158. Hence, it is suggested that MgIG bears potential activity against SIRT3.

### 2.6. SIRT3 Mediates the Protective Effect of MgIG on Mitochondrial Dysfunction Induced by Cisplatin

To further investigate whether the effects of MgIG depend on SIRT3, SIRT3 was knocked down in HK-2 cells with siRNA (Figure 6A,B), and the cytotoxicity and mitochondrial function were examined. In the negative control siRNA(si-NC) group, when compared with cisplatin alone, the combination with MgIG improved cell viability from 65.5% to 96.0% (Figure 6C), and the apoptosis rate decreased from 26.9% to 12.3% (Figure 6D,E). After SIRT3 knockdown, MgIG did not show the original protective effects, given that the cell viability again declined to 74.9% (Figure 6C) and the apoptosis rate increased to 27.8% (Figure 6D,E). The same tendency was observed for the mitochondrial membrane potential (Figure 6F,G). Compared with the si-NC group in combination with MgIG, silencing of SIRT3 increased the proportion of JC-1 monomer–positive HK-2 cells, suggesting that knockdown of SIRT3 attenuated the protective effect of MgIG on cisplatin-induced damage in HK-2 cells.

We also evaluated the effect of Honokiol (HKL), an agonist of SIRT3, on cisplatin-induced HK-2 cell viability (Figure 6H). With co-administration of HKL at 0.5, 1, and 2 μM, the cell viability increased in a dose-dependent manner, compared with the cisplatin group, while the inhibition of proliferation was not observed with HKL alone (Appendix A). In addition, HKL reduced the amount of platinum–mtDNA (Figure 6I) but had no effect on platinum–nDNA, a phenomenon identical to that shown for MgIG, suggesting that agonism of SIRT3 minimizes the amount of platinum bound to mtDNA.

## 3. Discussion

In this study, we evaluated the protective effect and possible mechanisms of MgIG against cisplatin-induced nephrotoxicity. We confirmed that cisplatin tended to accumulate in mitochondria, and MgIG reduced the concentration of cisplatin in mitochondria and target-binding to mtDNA. In vivo and in vitro, MgIG significantly ameliorated cisplatin-induced mitochondrial morphological abnormalities, dysfunction, and energy metabolism disorders, and reduced the levels of SIRT3. We further verified that MgIG played a protective role against cisplatin-induced mitochondrial dysfunction through SIRT3, and an agonist of SIRT3 was able to specifically decrease the amount of platinum–mtDNA adducts without affecting platinum–nDNA adducts.

Glycyrrhetinic acids are the predominant active substances in licorice, and natural glycyrrhetinic acids consist mainly of 18β-glycyrrhetinic acid and trace amounts of 18α-glycyrrhetinic acid [35]. It has been well-documented that 18β-glycyrrhetinic acid has a protective effect against various renal injuries [36,37,38], but 18α-glycyrrhetinic acid has not yet been analyzed. MgIG is a magnesium salt of 18α-glycyrrhetinic acid, and in this study, the effect of Mg ions was excluded, showing that the protective effect of MgIG is attributed to 18α-glycyrrhetinic acid.

The kidneys require a sufficient amount of ATP to transport water and solutes against a gradient for their core function. They contain the highest number of mitochondria per unit weight among all organs, excluding the heart. Since mitochondria produce more than 90% of ATP through oxidative phosphorylation, they are essential for maintaining normal renal function [39]. There is increasing evidence [12,13,40] suggesting that targeting mitochondria is a practical and feasible approach to prevent and treat kidney disease. Currently, the reported strategies include inhibition of the mitochondrial apoptotic pathway, prevention of mPTP pore opening, regulation of the kinetic process of mitochondrial fission/fusion, and development of antioxidants targeting mitochondria, some of which are being tested clinically under different clinical situations [12,13,40]. However, no drugs have been reported to ameliorate cisplatin-induced renal injury by reducing the concentration of cisplatin in mitochondria.

DNA is the main target of cisplatin, and cisplatin binds to DNA in the form of adducts, which are recognized and responded to by proteins, thereby triggering events such as ROS generation, oxidative stress, apoptosis, and ultimately cell death [41,42]. It has been reported [14,43] that cisplatin binds 2–7 times more to mtDNA than to nDNA in rat liver and brain. Due to the lack of nucleotide excision repair pathways in mitochondria, unrepaired platinum–mtDNA adducts persist [44], which interfere with mtDNA transcription, leading to reduced expression of components encoded by mtDNA in the electron transport chain (ETC), impaired respiratory capacity and, subsequently, ROS production [45]. The results of the present study and those of previous studies [46] provide supporting evidence for the accumulation of cisplatin in renal tubular mitochondria. In this study, MgIG co-administration remarkably reduced the concentration of cisplatin in mitochondria and specifically reduced the amount of platinum–mtDNA adducts while not affecting nDNA adducts, implying that MgIG exerts a protective effect primarily by targeting mitochondria.

Mitochondrial dysregulation plays a critical pathogenic role in cisplatin nephrotoxicity, especially in renal tubular cell injury and death. In the present study, MgIG markedly alleviated cisplatin-induced mitochondrial dysfunction, including mitochondrial ROS production, membrane potential depolarization, oxidative phosphorylation capacity, and ATP reduction. Mitochondria are one of the predominant endogenous sources of ROS. Marullo stated that [47] mtROS production was not related to the level of drug-induced nDNA damage but to mtDNA damage. It was further suggested that MgIG protects renal tubular cells from cisplatin by targeting mitochondria and reducing the binding amount of cisplatin to mtDNA, thereby reducing ROS in mitochondria.

Sequentially, MgIG dramatically ameliorated cisplatin-induced mitochondrial cristae fragmentation in HK-2 cells; in vivo experiments clearly showed the altered mitochondrial microstructure. Under physiological conditions, mitochondria undergo constant fission and fusion, and the dynamic balance between them is crucial for the function and viability of mitochondria. After toxic stimulation of the kidneys, the balance between fission and fusion favors fission, while excessive fission is generally described as mitochondrial fragmentation [11]. As reported [48], weakening mitochondrial fission or enhancing its fusion utilizing gene editing can protect mice from different AKI models.

Naturally, we wanted to find out how MgIG decreases platinum–mtDNA adducts to protect renal tubular cells. First, the effects of the uptake transporters OCT2 and CTR1 were excluded. Second, cisplatin is transported from the cytoplasm to the mitochondria by COX17; the increase of COX17 promotes the accumulation of platinum in the mitochondria, contributing to overall cisplatin cytotoxicity [49]. Still, there was no clear difference between the cisplatin group and MgIG + cisplatin group. Therefore, we speculate that the renoprotective effect of MgIG is not established by alterations of the transporters.

Due to high-energy consumption, renal tubular mitochondrial damage inevitably leads to multiple renal energy metabolism disorders. Therefore, we explored the underlying mechanisms from metabolomics. The results showed that amino acid metabolism, NAD^+^ metabolism, and redox substances were all changed, which supported the importance of mitochondrial dysfunction in renal disease. In AKI and CKD, renal NAD^+^ levels are reduced due to decreased biosynthesis and increased consumption [29]. Finally, we focused on NAD^+^ metabolism due to its critical role in renal disease.

NAD^+^, an electron carrier for glycolysis, Krebs cycle, and β-oxidation to ETC, is essential for efficient ATP production. Since most bioenergetic and NAD^+^-dependent signaling pathways occur in distinct subcellular compartments, these metabolic processes require locally accessible NAD^+^ [50]. Mitochondria contain the highest levels of NAD^+^ among the subcellular organelles, with a concentration of approximately 250 μM [51]. Maintaining mitochondrial NAD^+^ levels is essential for cell survival, especially when NAD^+^ pools in nuclei and cytoplasm are depleted under stressful conditions [52]. In addition to acting as an electron carrier, NAD^+^ can act as a substrate for signaling enzymes that regulate various signaling pathways, including poly ADP ribose polymerase (PARP), sirtuins, and ectonucleotidases [30]. Since MgIG targets mitochondria, we next focused on mitochondrial NAD^+^-regulated enzymes. Among the three enzymes above, PARP is mainly located in the nucleus; ectonucleotidases are membrane proteins; and only three isoforms of sirtuins (SIRT3, SIRT4, and SIRT5) are in the mitochondrial matrix [50,53]. Van de Ven et al. [54] used a proteomic approach to study mitochondrial SIRT3, SIRT4, and SIRT5 protein–protein interactions and found that most proteins interacted with SIRT3. SIRT3 is related to proteins involved in amino acid metabolism, fatty acid oxidation, the TCA cycle, and the ETC/OXPHOS complex, which also highlights the central role of SIRT3 in mitochondrial metabolism [53]. We showed that cisplatin caused decreases in SIRT3, SIRT4, and SIRT5 protein expression levels, and MgIG had the most pronounced back-regulatory effect on SIRT3. According to the molecular docking experiments, MgIG binds to amino acids at several SIRT3 sites, suggesting potential activity of MgIG on SIRT3.

We demonstrated that MgIG played a partial protective role in an SIRT3-dependent manner by knocking down SIRT3. According to Yang [54], SIRT3 is necessary for restoring membrane potential and contributes to membrane integrity. Our data also showed that MgIG restored mitochondrial membrane potential and maintained membrane integrity by regulating SIRT3 level in HK-2 cells. Morigi [55] proposed that cisplatin-induced dysfunction in proximal renal tubular cells results from a decrease in SIRT3, which leads to the recruitment of fission protein Drp1 on the mitochondrial membrane, and pro-fusion kinetic-related protein OPA1 downregulation, ultimately pushing mitochondrial dynamics toward division. This is in agreement with the mitochondrial morphology and ultrastructure observed in our study, which suggests that MgIG optimizes mitochondrial fission–fusion kinetics by regulating SIRT3.

In addition, SIRT3 promotes mtDNA repair [56] and improves cellular resilience to environmental stress [57]. PGC1α and SIRT3 form a positive feedback loop that jointly regulates mitochondrial energy metabolism [58]. Several recent studies also support the view regarding the renoprotective activity of SIRT3. For instance, curcumin prevents alterations in mitochondrial ultrastructure, energy production, redox homeostasis, and kinetics in AKI mice by modulating SIRT3 levels [59]. HKL [60], resveratrol [61], and silymarin [62], all of which are potential SIRT3 agonists, have also been demonstrated to have nephroprotective effects.

Up to now, there has been no published evidence on the relationship between SIRT3 and the amount of platinum–mtDNA adducts. However, we found that combination with HKL, an agonist of SIRT3, substantially reduced the amount of platinum–mtDNA adducts, similar to MgIG, without affecting the binding content of platinum to nDNA. Therefore, it is inferred that the renoprotective mechanism of MgIG involves increasing the level of SIRT3, thus reducing mtDNA adduct levels and mitigating mitochondrial damage. To analyze why the level of SIRT3 affects platinum–mtDNA, we considered two points. On the one hand, we took into account that SIRT3 is mitochondria-specific, can stimulate mitochondrial biogenesis, and also interacts with PGC-1α to produce more mtDNA [58]. We found that MgIG elevated the level of SIRT3, so it can be speculated that MgIG reduces mitochondrial damage by increasing the total amount of mtDNA and mitigating the mtDNA damage. On the other hand, it may be related to the deacetylation function of SIRT3. It has been stated [63] that acetylation modification can neutralize the positive charge carried by protein lysine residues and weaken the interaction between histones and DNA, which facilitates the release of DNA from histones and allows an easy attack by cisplatin. Thus, when inhibiting intracellular deacetylase, the acetylation of histones can be increased, and the intracellular DNA–cisplatin adducts are increased. Although mtDNA is not protected by histones, there may still be a similar mechanism that deserves further investigation.

In conclusion, MgIG significantly ameliorated cisplatin-induced mitochondrial morphological and functional abnormalities as well as NAD^+^ metabolism disorders of the kidneys both in vivo and in vitro. MgIG can bind to SIRT3 and prevent the decreased expression of SIRT3 induced by cisplatin. Furthermore, SIRT3 affects the binding amount of cisplatin to mtDNA and mediates the protective effect of MgIG on mitochondrial dysfunction (Figure 7). Our study provides a new therapeutic strategy for the treatment of cisplatin-induced nephrotoxicity.

## 4. Materials and Methods

### 4.1. Chemicals and Materials

Cisplatin (Platinum content > 98%) was purchased from Shanghai Yuanye Biotechnology Co. (Shanghai, China); MgIG Injections were provided by Chia Tai Tianqing Pharmaceutical Group Co. (Lianyungang, Jiangsu, China); MgIG standard was obtained from National Institutes for Food and Drug Control (Beijing, China); oleanolic acid, NAD^+^, NADH, NADPH, NR, NMN, NAM, 2-chloroadenosine were bought from Sigma-Aldrich (St Louis, MO, USA). Penicillin-streptomycin, DMEM-F12, and trypsin were acquired from Gibco (Grand Island, NY, USA); Fetal bovine serum (FBS) was procured from Genetimes Biotechnology Co. (Shanghai, China); PCR reaction primers were purchased from Invitrogen (Shanghai, China). Honokiol (HKL) was purchased from Selleck (Shanghai, China). The remaining reagents were commercially available.

### 4.2. Animal Experiments

The male C57BL/6J mice (8 weeks old, weighing 20–22 g) were obtained from Zhejiang Vital River Laboratory Animal Technology Co. (Jiaxing, China). They were kept under a 12 h light/dark cycle at about 25 °C with free access to pathogen-free food and purified water. All animal experiments were strictly performed in accordance with the Guide for the Care and Use of Laboratory Animals and designed to minimize suffering and reduce the number of animals used. Every effort was made to alleviate animal pain, suffering, and distress and to reduce the number of animals used.

After one week of adaptation, 24 mice were randomly divided into 3 groups (*n* = 8). (1) Control group: vehicle aqueous saline ip administration once daily; (2) Cisplatin group: 3.5 mg/kg cisplatin ip administration once daily for the first 4 d of each cycle; (3) MgIG + Cisplatin group: 15 mg/kg MgIG ip administration once daily, 3.5 mg/kg cisplatin administration as cisplatin group. The experiment was performed in 3 cycles. In cycle 2 and cycle 3, from d −2 to d 8, each mouse was injected subcutaneously 1 mL saline to hydrate for 10 consecutive days. 

Mice were anesthetized with isoflurane and euthanized by spinal dislocation, and then the kidneys and serum were obtained. The kidneys were separated, two portions were preserved in 4% paraformaldehyde or 3% glutaraldehyde for subsequent histopathology, the others and the serum were frozen at −80 °C.

### 4.3. Creatinine and Urea Nitrogen Determination

The creatinine and urea nitrogen in serums were determined using colorimetry method as described in Creatinine assay kit and Urea nitrogen assay kit (Jiancheng, Nanjing, Jiangsu, China).

### 4.4. Histopathology Assessment

The Pathology and PDX Efficacy Evaluation Center of China Pharmaceutical University provided the kidneys tissue sections and staining with hematoxylin and eosin (H&E). For transmission electron microscopy, samples of the renal cortex were prepared according to the methods that we have described previously [64].

### 4.5. Cell Culture and Administration

The HK-2 cells were obtained from Shanghai Zhong Qiao Xin Zhou Biotechnology Co. and cultured in DMEM/F12 medium supplemented with 10% FBS and 1% penicillin-streptomycin. The culture condition was maintained in 5% CO_2_ and 95% air atmosphere at 37 °C. The cells in the logarithmic growth phase were seeded, divided into 4 groups and administrated as follow: Control (vehicle aqueous saline), Cisplatin (cisplatin 20 μM), MgIG + Cisplatin (cisplatin 20 μM and MgIG 100 μM) and MgIG (MgIG 100 μM). 

### 4.6. Subcellular Fractionations and DNA Isolation

The cells were cultured in 10 cm dishes and treated as described earlier. After harvest and rinse, the Mitochondria/Nuclei Isolation Kit (KGA828, KeyGEN BioTECH, Nanjing, Jiangsu, China) was performed to separate the nucleus and mitochondria from the cytoplasm, as described previously [65]. Instructions from the TaKaRa MiniBEST Universal Genomic DNA Extraction Kit (TaKaRa, Dalian, Liaoning, China) were followed, and DNA from mitochondria and cytoplasm were isolated.

### 4.7. Platinum Concentration Assay

In this experiment, the platinum concentration was used to characterize the amount of cisplatin [66]. Cells in 12-well plates, isolated subcellular organelles or DNA were collected using 150 µL ddH2O for each sample and ultrasonic fragmentation, added 30 µL nitric acid. After digestion at 60 °C for 1h, the mixture was centrifuged at 12,000× *g* for 15 min to obtain the supernatant. The concentration of platinum was then measured by ICP-MS (NexION 350D, Perkin Elmer, Waltham, MA, USA) as described previously [67], and corrected by protein level. 

### 4.8. MgIG Concentration Assay

Pipette 60 μL from each sample, add 6 μL of 100 ng/mL oleanolic acid (internal standard, IS) and 300 μL of methanol. After vortex and centrifugation at 18,000× *g* for 10 min, the supernatant was analyzed by LC-MS/MS with a Shimadzu HPLC system (Kyoto, Japan) and an API 4000 triple quadrupole mass spectrometer (Framingham, MA, USA). Chromatographic separation was conducted on a ZORBAX Eclipse Plus C18 column (4.6 × 150 mm, 5 µm). Mobile phase A was an aqueous solution containing 0.1% formic acid and 2 mM amine acetate, and mobile phase B was acetonitrile. Gradient elution: 0–0.5 min, 30% (*v*/*v*) B; 2.5 min, 95% B; 8 min, 95% B; 9 min, 30% B; 10 min, stop analysis; flow rate: 0.7 mL/min; column temperature: 40 °C. MgIG and oleanolic acid were monitored severally using the MRM transitions to form the [M+H]^+^ precursor ions of *m*/*z* 469.4→425.4 and *m*/*z* 455.5→455.5 2. The declustering potential (DP) and collision energy (CE) for MgIG and oleanolic acid were −60 V and −50 V, 45 eV and 32 eV, respectively. Data acquisition and analysis were performed using Analyst TF 1.5.1 software (SCIEX, Framingham, MA, USA).

### 4.9. JC-1 Mitochondrial Membrane Potential Detection

The culture medium was replaced with a sufficient JC-1 working solution (Beyotime, Shanghai, China). Next, the cells were incubated at 37 °C for 15 min, and washed with PBS. Subsequently, the ratio of green fluorescence was detected using BD Accuri C6 flow cytometry (BD biosciences, Franklin Lake, NJ, USA). 

### 4.10. MitoSOX Mitochondrial Superoxide Staining

The culture medium was removed, and a 5 μM MitoSOX working solution was applied (Invitrogen, Carlsbad, CA, USA) to cover the cells, and subsequently incubated for 10 min at 37 °C in the dark. The cells were then washed with PBS and imaged using a Lionheart Fx automated microscope (Bio-Tek, Winooski, VT, USA).

### 4.11. MitoTracker^®^ Deep Red FM Staining

MitoTracker^®^ Deep Red FM is used for mitochondrial localization. After cells culture, the medium was discarded, and a 500 nM MitoTracker^®^ Deep Red FM working solution (Invitrogen, Carlsbad, CA, USA) was added and incubated for 20 min at 37 °C. Then washed and fixed with 4% paraformaldehyde, the cells were stained with Hoechst 33,342 (Beyotime, Shanghai, China). Finally, FV3000 confocal microscopy (Olympus, Japan) was used for observation.

### 4.12. Mitochondria Cellular Respiration and Intracellular ATP Measurement

Following the manufacturer’s instructions, oxygen consumption rates (OCR) were measured in real-time with the Mito Stress Test Kit (Agilent, California, Santa Clara, CA, USA), using the Seahorse XFe96 Analyzer (Agilent, California, CA, USA), intracellular ATP levels were determined using an enhanced ATP assay kit (Beyotime, Shanghai, China). Data were normalized by protein concentration.

### 4.13. Cell Viability and Apoptosis 

According to the manufacturer’s protocol, the cell viability and apoptosis were tested by using Cell Counting Kit-8 (CCK-8) (Beyotime, Shanghai, China) and Annexin V-FITC Apoptosis Detection Kit (Beyotime, Shanghai, China), respectively. 

### 4.14. Metabolomics Studies of Kidney

As reported previously, the GC/MS and LC/Q-TOF MS metabolomics analysis methods were performed [68,69]. For the obtained multivariate data, partial least squares discriminant analysis (PLS-DA) and metabolomics pathway analysis of the differential compounds were conducted using MetaboAnalyst (www.metabonalyst.ca (accessed on 6 June 2021)) [70].

### 4.15. Nicotinamide Adenine Dinucleotide (NAD^+^)-Related Substance Examination

NAD^+^ and NADH were detected using an NAD^+^/NADH detection kit (Beyotime, Shanghai, China). Other substances are determined as follows: kidney samples were processed as for metabolomics assay, cells were collected and fragmented as for the MgIG concentration assay. For IS, 2-Chloroadenosine was used. A Shimadzu HPLC system (Kyoto, Japan) and a SCIEX Triple Quad^TM^ 5500 mass spectrometer (Framingham, MA, USA) were applied. Chromatographic separation was conducted on a SiELC Obelisc R column (2.1 × 150 mm, 5 µm). Mobile phase A was an aqueous solution containing 0.1% formic acid and 7.5 mM amine acetate, and mobile phase B was acetonitrile. Gradient elution was as follow: 0–0.5 min, 90% (*v*/*v*) B; 2.5 min, 90% B; 6 min, 10% B; 11 min, 10% B; 15 min, 90% B; 15 min, stop analysis; flow rate: 0.3 mL/min; column temperature: 40 °C. In positive ion mode, the temperature of the source was maintained at 550 °C. The ion injection voltage is 4500 V. In negative ion mode, the temperature of the source was maintained at 500 °C, and the ion injection voltage is −4500 V. The detected ion pairs of substances are shown in Appendix A. Data acquisition and analysis were performed using Analyst TF 1.5.1 software (SCIEX, Concord, ON, Canada).

### 4.16. Quantitative Real-Time PCR Analysis

Total RNA extraction was performed according to the RNAiso Plus reagent (Takara, Dalian, Liaoning, China) protocol. Reverse transcription of 500 ng of total RNA to evaluate genes was performed using HiScript III-RT SuperMix (Vazyme, Nanjing, Jiangsu, China); Quantitative reverse transcription PCR analysis was performed using SYBR Green PCR Master Mix and a MyIQ real-time PCR cycler (Bio-Rad, Hercules, CA, USA). The expression of the targets was normalized to that of β-actin. The primer sequences are listed in Appendix A.

### 4.17. Western Blot

Cells and mice tissue were lysed in RIPA buffer with PMSF (Beyotime, Shanghai, China), and then protein was adjusted to the same level, added loading buffer and boiled at 100 °C for 5 min. A 50µg mass of protein lysates were loaded on 10% SDS-PAGE gels and transferred using PVDF membranes. After block, the membranes were incubated overnight with anti-SIRT3 (1:1000; Thermo, Waltham, MA, USA), anti-SIRT4 (1:1000; Thermo, Waltham, MA, USA), anti-SIRT5 (1:1000; Proteintech, Wuhan, Hubei, China) or anti-β-actin primary antibodies (1:1000; Cell Signaling Technology, Danvers, MA, USA), followed by incubation with appropriate secondary antibodies (1:10,000; Cell Signaling Technology, MA, USA) at 37 °C for 1h. The signal was visualized and captured with a ChemiDoc XRS+system (Bio-Rad, Hercules, CA, USA). 

### 4.18. Transfection

According to the previously described methods [71], the HK-2 cells were transfected with SIRT3-siRNA or NC (Negative control)-siRNA (GenePharm, Shanghai, China), and Lipofectamine^TM^ RNAiMAX reagent (Invitrogen, Carlsbad, CA, USA).

### 4.19. Molecular Docking

The 2D chemical structure of MgIG was obtained from the PubChem database (https://pubchem.ncbi.nlm.nih.gov/ (accessed on 20 October 2021)), and the 3D structure of SIRT3 (PDB ID: 4JSR) was fetched from the PDB database (https://www.rcsb.org/ (accessed on 21 October 2021)). Molecular docking was performed via Maestro (Schrödinger, Inc., New York, NY, USA) and finally visualized by Pymol (Schrödinger, Inc., New York, NY, USA).

### 4.20. Statistics

All data are presented as the mean ± SD. An unpaired t-test with equal variance and one-way or two-way ANOVA followed by Sidak’s multiple comparisons test was used for statistical analyses. Statistical data analysis was performed using GraphPad Prism 8.1. Differences were considered significant at *p* < 0.05.

## Figures and Tables

**Figure 1 ijms-23-13093-f001:**
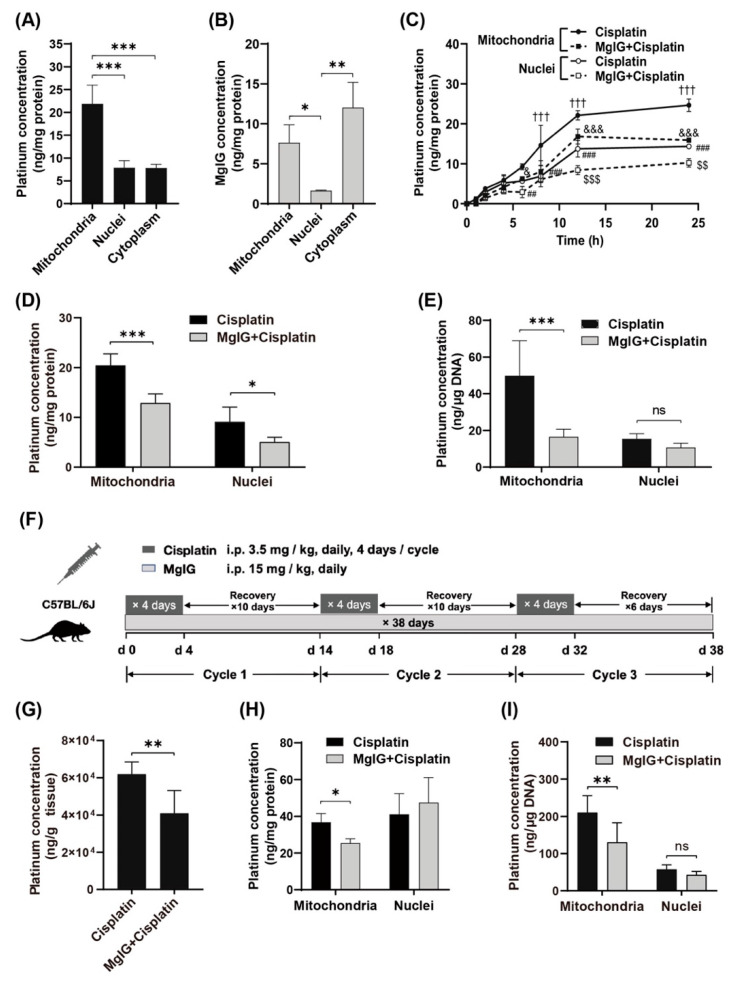
MgIG reduces the concentration of cisplatin in mitochondria and target-binding amount to mtDNA. (**A**) The subcellular distribution of platinum in HK-2 cells, *n* = 5. Cisplatin tends to accumulate in mitochondria. (**B**) The subcellular distribution of MgIG in HK-2 cells, *n* = 3. MgIG amount in mitochondria is higher than that in nuclei. (**C**) The concentration–time process of platinum in mitochondria and nuclei of HK-2 cells, *n* = 3. (**D**) The subcellular distribution of platinum after MgIG combination in HK-2 cells for 24 h, *n* = 5. Co-treatment with MgIG reduces the accumulation of cisplatin in mitochondria. (**E**) The effect of MgIG on the concentration of platinum bound to DNA, *n* = 5. MgIG specifically reduces platinum–mtDNA level. (**F**) Flow chart of administration to mice in the designated groups. (**G**) The platinum concentration in kidneys, *n* = 6. Combination with MgIG reduced platinum concentration in kidneys. (**H**,**I**) The concentration of platinum and its binding amount to DNA in mitochondria and nuclei of kidneys. MgIG reduced the concentration of platinum in mitochondria and its binding to DNA in kidneys. The data are presented as the mean  ±  SD. * *p* < 0.05, ** *p* < 0.01, *** *p* < 0.001,ns indicates no significant difference; ††† *p* < 0.001 the Mitochondria–cisplatin group versus the Mitochondria–MgIG + cisplatin group; & *p*  <  0.05, &&& *p* < 0.001 the Nuclei–MgIG + cisplatin group versus the Mitochondria–MgIG + cisplatin group; ## *p*  < 0.01, ### *p* < 0.001 the Mitochondria–cisplatin group versus the Mitochondria–cisplatin group; and $$ *p* < 0.01, $$$ *p* < 0.001 the Nuclei–cisplatin- group versus the Nuclei–MgIG + cisplatin group.

**Figure 2 ijms-23-13093-f002:**
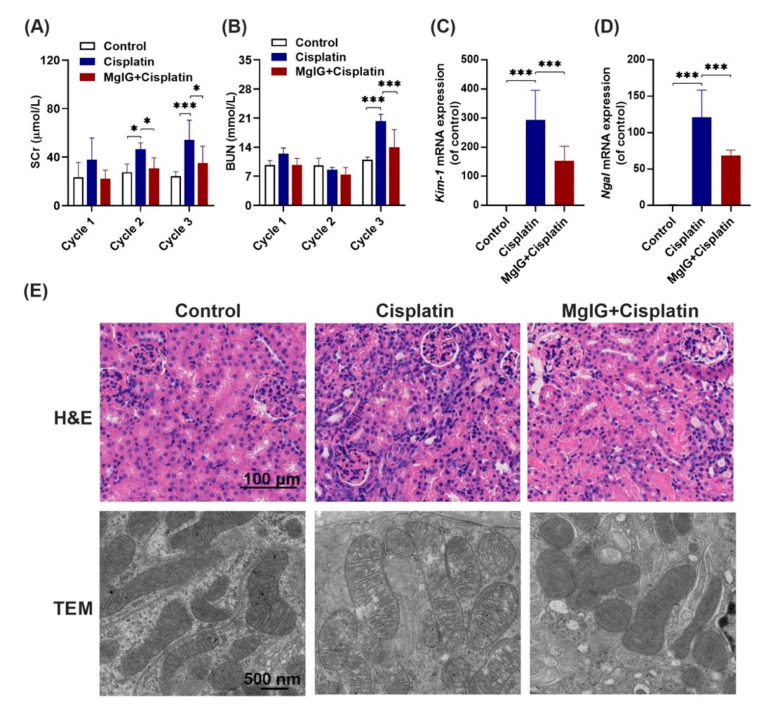
MgIG significantly ameliorates cisplatin-induced renal injury. (**A**,**B**) The levels of SCr and BUN in mice over 3 cycles. (**C**,**D**) The mRNA expression levels of KIM-1 and NGAL in mouse kidneys at the end of cycle 3. Combination with MgIG attenuated the elevation of SCr, BUN, KIM-1, and NGAL caused by cisplatin in mice. (**E**) Representative H&E staining of renal mouse tissue (scale bar = 100 μm) and TEM micrographs of renal tubular epithelial cell mitochondria (scale bar = 500 nm) from each group. Co-treatment with MgIG prevented mitochondrial ultrastructural abnormalities in renal tubules against cisplatin. The data are presented as the mean  ±  SD, *n* = 8. * *p* < 0.05, *** *p* < 0.001.

**Figure 3 ijms-23-13093-f003:**
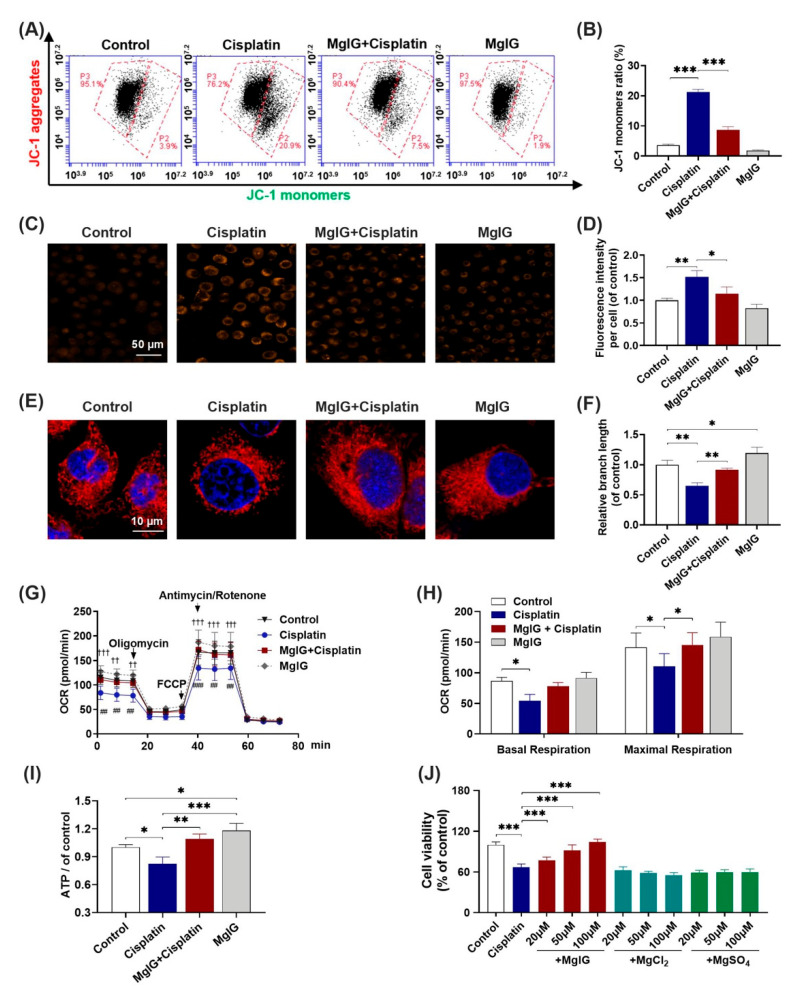
MgIG notably alleviates cisplatin-induced mitochondrial dysfunction. (**A**,**B**) The representative images of JC-1 staining of HK-2 cells evaluated by a flow cytometer, *n* = 4. (**C**,**D**) Representative images showing the mitochondrial ROS of HK-2 cells after staining with MitoSOX and quantitative analysis, *n* = 3. (Scale bar = 50 μm). (**E**,**F**) Representative images showing the mitochondrial morphology of HK-2 cells after staining with MitoTracker^®^ Deep Red and quantitative results, *n* = 3. (Scale bar = 10 μm.) (**G**,**H**) Representative traces of the measurement of oxygen consumption rate (OCR), *n* = 5. Oligomycin, FCCP, and antimycin/rotenone were administrated at the indicated time points. The production of basal and maximal respiration was determined according to the OCR values. Co-incubation with MgIG suppressed cisplatin-induced depolarization of the mitochondrial membrane potential, mtROS production, OCR decrease, and morphological abnormalities. †† *p* < 0.01, ††† *p* < 0.001, the cisplatin group versus the control group. ## *p* < 0.01, ### *p* < 0.001, the cisplatin group versus the MgIG + cisplatin group. (**I**) The intracellular ATP level. MgIG co-treatment reversed cisplatin-induced ATP production decrease in HK-2 cells. (**J**) The cell viability evaluated by CCK-8, *n* = 5. Mg salts offered no protection from proliferation inhibition caused by cisplatin in HK-2 cells. The data are presented as the mean ± SD. * *p* < 0.05, ** *p* < 0.01, *** *p* < 0.001.

**Figure 4 ijms-23-13093-f004:**
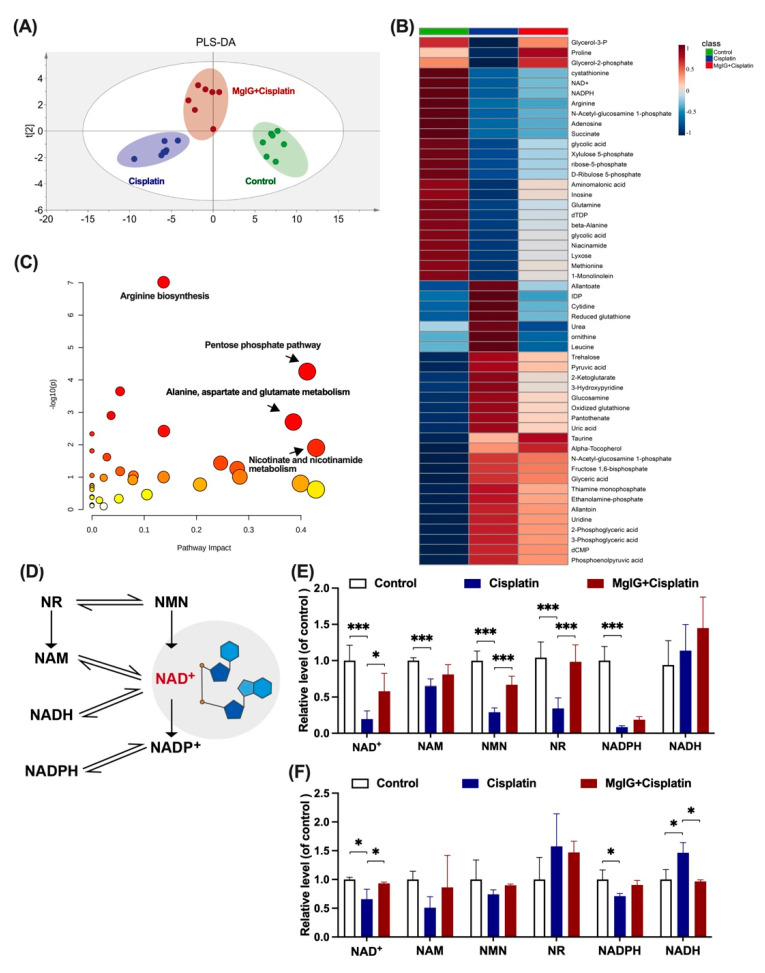
MgIG markedly ameliorates cisplatin-induced energy metabolism disorder and recovers NAD^+^ biosynthesis. (**A**) Partial Least Squares Discrimination Analysis (PLS-DA) plot depicting the differences in kidneys from three assigned groups, *n* = 7. There is a clear distinction between the groups. (**B**,**C**) Heat maps and KEGG pathway analysis of differential compounds in kidneys from the three assigned groups. Combination with MgIG improved cisplatin-induced energy metabolism disorder. Red circles represent high *p* value, yellow circles represent low *p* value, the sizes of the circles are proportional to the pathway impact, *n* = 7. (**D**) The main biosynthesis pathways of NAD^+^. (**E**) The levels of NAD^+^, NADH, NAM, NMN, NR, and NADPH in the kidneys, *n* = 5. (**F**) The levels of NAD^+^, NADH, NAM, NMN, NR, and NADPH of HK-2 cells in the assigned groups, *n* = 3. Combination with MgIG increased the reduction of NAD+ and related substances stimulated by cisplatin. The data are presented as the mean ± SD. * *p* < 0.05, *** *p* < 0.001.

**Figure 5 ijms-23-13093-f005:**
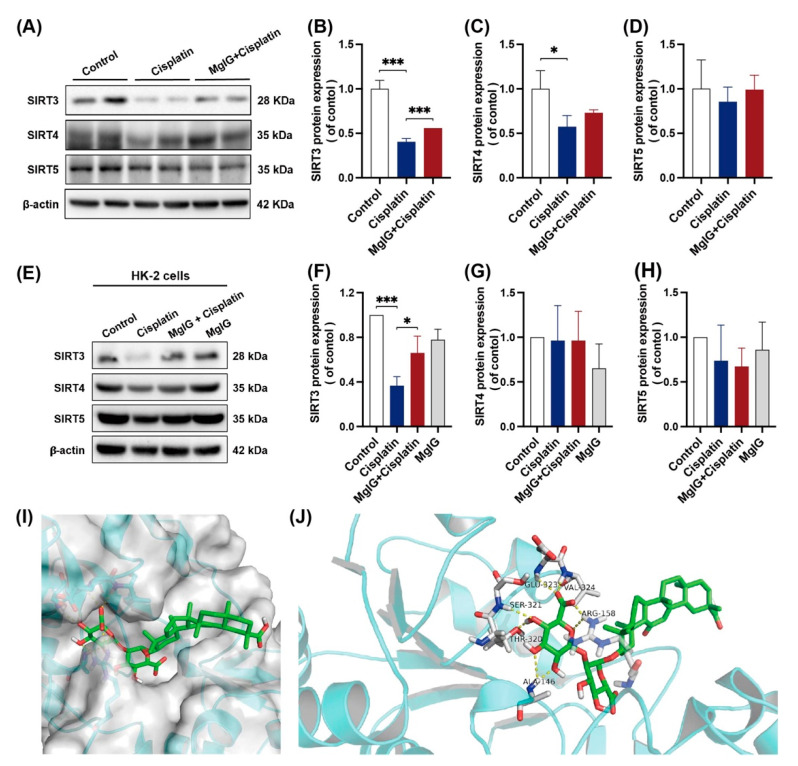
MgIG can bind to SIRT3 and prevent the decreased expression of SIRT3 induced by cisplatin. (**A**–**D**) Representative images and quantitative results of Western blot showing SIRT3, SIRT4, and SIRT5 expression in kidneys from the indicated groups, *n* = 3. (**E**–**H**) Representative images and quantitative results of Western blot showing SIRT3, SIRT4, and SIRT5 expression in HK-2 cells, *n* = 3. After co-treatment with MgIG, the SIRT3 level in both HK-2 cells and kidneys was upregulated. (**I**,**J**) Molecular docking results showing the possible sites of MgIG acting with SIRT3. MgIG could form hydrogen bonds with ALA at 146, ARG at 158, THR at 320, SER at 321, GLU at 323, and VAL at 324 of SIRT3, and π–π interaction with ARG at 158. The data are presented as the mean ± SD, *n* = 3. * *p* < 0.05, *** *p* < 0.001.

**Figure 6 ijms-23-13093-f006:**
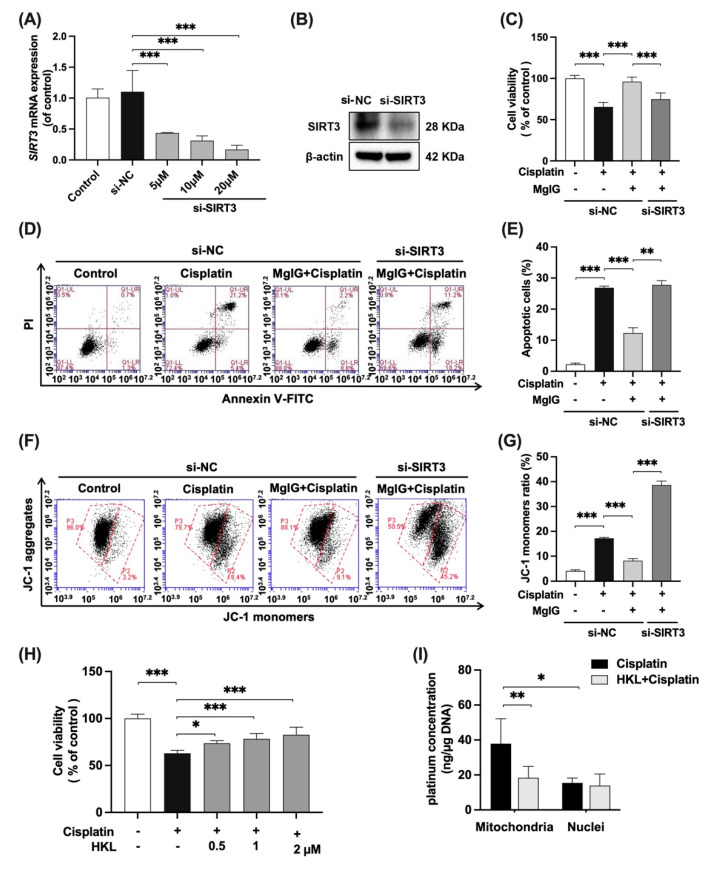
SIRT3 mediates the protective effect of MgIG on mitochondrial dysfunction induced by cisplatin. SIRT3 was knocked down in HK-2 cells by transferring siRNA. (**A**,**B**) The mRNA and protein expression levels of SIRT3 in HK-2 cells, *n* = 3. (**C**–**G**) The cell viability, representative images of apoptosis, and JC-1 staining of HK-2 cells from the indicated groups, comparing SIRT3 knockdown with the si-NC group, and quantitative results, *n* = 4. SIRT3 knockdown weakened the protective effect of MgIG combination from cisplatin in HK-2 cells. (**H**) The cell viability of HK-2 cells treated with cisplatin and HKL, *n* = 5. HKL mitigated the proliferation inhibition of HK-2 cells against cisplatin. (**I**) The effect of HKL on the amount of platinum–mtDNA adducts and platinum–nDNA adducts, *n* = 5. HKL specifically reduced platinum–mtDNA level. The data are presented as the mean ± SD. * *p* < 0.05, ** *p* < 0.01, *** *p* < 0.001.

**Figure 7 ijms-23-13093-f007:**
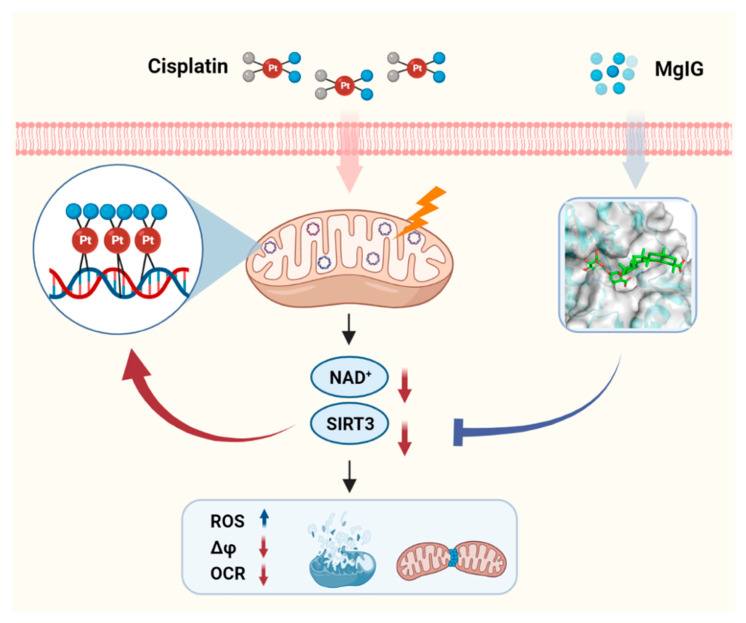
Schematic depicting the role of MgIG in cisplatin-induced kidney injury. Cisplatin tends to accumulate in the mitochondria of renal tubular epithelial cells and binds to mtDNA, causing a decrease in NAD+ and SIRT3, resulting in mitochondrial dysfunction and cytotoxicity. MgIG can bind to SIRT3 and reduce the target-binding amount of cisplatin to mtDNA, thus exerting a protective effect on cisplatin-induced renal injury through SIRT3.

## Data Availability

The data contained within the paper are available from the authors upon request.

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
