# Peer review of "Magnesium Isoglycyrrhizinate Reduces the Target-Binding Amount of Cisplatin to Mitochondrial DNA and Renal Injury through SIRT3"

_ijms, 2022, doi:10.3390/ijms232113093_

Round 1
Reviewer 1 Report
The study by Wang and co-workers was to determine the mechanisms and signaling pathways of Magnesium isoglycyrrhizinate (MgIG) for kidney protection in cisplatin-induced kidney injury. By using in vivo mouse model and in vitro cultured HK-cell studies, the authors showed that cisplatin-induced nephrotoxicity was due to the accumulation of the platinum binding to mitochondrial DNA rather than nuclear DNA in renal tubular epithelial cells and hence elicited mitochondrial morphological alteration and dysfunction. Co-treatment of MgIG attenuated cisplatin-induced mitochondrial dysfunction via preventing downregulated NAD+-dependent deacetylase sirtuin-3 (SIRT3) pathway. Thus, the study indicates that the amount of mtDNA is crucial in cisplatin-induced kidney injury. The study is well designed and written, and scientifically relevant to clinical application. The topic is fit to the journal scope. However, there are some concerns that need to be addressed by the authors.
Major concerns and suggestions
· Although it was a well-designed study, at least a hypothesis should be given based on the observations.
· Was there a reason to perform the study on Day 6 after the third cycle rather than 10 days?
· Was MgIG in a form of powder or liquid? How was MgIG prepared for the IP injection?
· The authors stated that each mouse received 1 ml saline subcutaneously to prevent dehydration for 10 consecutive days during the cycle 2 and 3. It seems that both cisplatin-treated alone and a combination of cisplatin and MgIG treated mice had some issues for not taking food and water properly. Did you monitor urine output, food and water consumption, body weight, or other physiological parameters which are also important markers determining the impact of MgIG against cisplatin-nephrotoxicity?
· The authors added a conclusion at the end of the main text. It would be better to move the conclusion section to the end of the discussion section.
· While the schematic picture is very helpful, it would be better to provide a thorough description of the figures.
Mirror concerns
· It is unclear in Figure 4 legend. Figure 4D does not show the levels of NAD+, NADH, NAM, NMN, NR, and NADPH in mouse kidneys. Maybe there are several incorrect labels in Figure 4D-4F.
· In Figure 6F, the font size of the numbers inside those figures was too small to be readable.
· The order of Table S1 and Table S2 should be switched in the supplemental materials. Typically, the numbers of figures or tables should be consecutive (i.e., no 2 then 1).
· Was Honokiol (HKL) purchased or gotten as a gift from Selleck?
· Line 190, the abbreviation of OCR should stand for oxygen consumption rate.
· A few grammar errors are found, such as mice kidneys, a sentence in Line 615, and others. A comma should be used after HKL instead of a period.
Reviewer 2 Report
The authors present some interesting findings and ideas about preventing cisplatin-induced nephrotoxicity. Studies are conducted in vivo in mice and in vitro in an immortalized cell line derived from human kidneys. The major concern with this work is the general lack of rigor in experimental design, model validation, and data interpretation. The authors have not sufficiently justified either their in vivo or their vitro model and key experiments lack critical controls.
Specific Comments:
1. From just reading the Abstract, an immediate question that comes to mind is: Does SIRT3 knockdown by itself cause mitochondrial dysfunction and cytotoxicity?
2. The Y-axes in Supplement Fig. 1 all contain a typographical error: "expression" is misspelled.
3. Line 53: Something missing here: What does “in the renal tubular” mean?
4. Lines 67-68: What does “mitochondrial enrichment properties” mean? This is not a standard way of describing an agent that improves mitochondrial function.
5. Lines 72-74: On what is the low-dose, repeated administration cisplatin mouse model based? A citation would seem to be needed if this is based on prior work that established the validity of the treatment approach to mimic clinical conditions.
6. Line 85: What is a “flat change”? I would presume you mean that there was no change.
7. Line 119: There are several places in the text, such as this line, where “mice” is improperly used and “mouse” should be used instead.
8. It is unclear whether the use of HK-2 cells is a valid in vitro model for this study. For one thing, if OCT2 is not expressed in HK-2 cells, how can this immortalized cell line be a valid model in which to study Cisplatin toxicity when this carrier is the major transporter for uptake of cisplatin into proximal tubular cells?
9. Section 2.2: SCr and BUN and other markers such as KIM-1 and NGAL are only indirect assessments of renal function. You have not directly assessed any aspect of renal function.
10. Line 272: The word “pointedly” is not correct here.
11. Lines 303-306: Not sure I really agree with this conclusion. The authors have not excluded that knockdown of SIRT3 is cytotoxic by itself. The data do not show the effect of SIRT3 knockdown without addition of MgIG.
12. The experiment with the SIRT3 agonist HKL is also problematic because there is no experiment with only HKL and no Cisplatin.
13. Line 356: This is not necessarily true; it may be just that the repair pathways are less efficient in mitochondria than in nuclei.
14. Line 482: Mice were not "executed," they were euthanized. it is also necessary to state what anesthetic agent was used and to describe the method of euthanasia.
Round 2
Reviewer 2 Report
The authors have adequately addressed the concerns regarding rationale and validation of the experimental models. There remain some grammatical and language errors that require correction.
Specific Comments:
1. L26: Change “mice” to “a mouse.”
2. L28: Change "exhibited" to "showed."
3. L51: Rewrite phrase "and so on."
4. L66: Change "And MgIG" to "MgIG also." Sentences should not begin with “And.”
5. L134: Change "mice" to "mouse."
6. L160: change "mice" to "mouse."
7. L166: The word "Furthermore" is not correct here.
8. L234: The word "differential" is not correct here. Do you just mean "different"?
9. L305: Change "raised" to "increased."
10. L313: Change "appeared to increase" to "increased."
11. L314-315: This sentence makes little sense. First, do not start a sentence with "And." The phrase "proliferation toxicity" is the problem. Do you just mean that "the inhibition of proliferation"?
12. L350: Evidence does not "grow." Change to "increasing amounts of."
13. L356: These drugs do not "facilitate cisplatin-induced renal injury" but prevent it or diminish it.
14. L390: Change "cytosol" to "cytoplasm."
15. L427: Move the reference citation "[54]" before "proposed."
